# Analysis of the Association between Metabolic Syndrome and Renal Function in Middle-Aged Patients with Diabetes

**DOI:** 10.3390/ijerph191811832

**Published:** 2022-09-19

**Authors:** Yoonjin Park, Su Jung Lee

**Affiliations:** 1Department of Nursing, Joongbu University, Geumsan-gun 32713, Korea; 2School of Nursing, Research Institute of Nursing Science, Hallym University, Chuncheon-si 24252, Korea

**Keywords:** metabolic syndrome, diabetes mellitus, renal function, glomerular filtration rate

## Abstract

This study investigated the effects of metabolic syndrome on the estimated glomerular filtration rate in middle-aged participants with diabetes to provide basic data to enable the development of education programs for middle-aged people to prevent diabetic kidney disease. This cross-sectional descriptive study analyzed data obtained in the 2nd year of the 8th Korea National Health and Nutrition Examination Survey in 2020 and enrolled 279 participants aged 40–65 years who were diagnosed with diabetes. Multilevel stratified cluster sampling was used to improve the representativeness of the samples and the accuracy of parameter estimation. The risk factors of metabolic syndrome and the risk of elevated eGFR were analyzed using regression analysis and the correlation between the variables was determined using Pearson’s correlation analysis. Middle-aged participants with diabetes whose eGFR was <90 showed a significant difference in their risk for metabolic syndrome based on sex, age, disease duration, and total cholesterol concentrations. Systolic blood pressure and waist circumference in men, and waist circumference and HDL cholesterol level in women were identified as risk factors that contribute to the increasing prevalence of metabolic syndrome.

## 1. Introduction

Changes in eating habits and lifestyle have contributed to the continued increase in the number of patients with obesity, and the number of patients with diabetes is increasing accordingly. The total number of registered patients with diabetes in 2020 indicates a very high prevalence of diabetes (10.7%); however, the glycemic control rate, which is defined as a glycated hemoglobin (HbA1c) level of <6.5, was only 24.1% [1]. Following Mexico and Turkey, South Korea has the third highest diabetes-related mortality rate among the countries of the Organization for Economic Co-operation and Development [2]. Therefore, national-level management of diabetes is very important. Moreover, the prevalence of diabetes is gradually increasing and is estimated to reach approximately 783 million by 2045 [3]. Furthermore, 28.6% of patients with type 2 diabetes have macrovascular complications, such as cardiovascular disease and peripheral arterial disease, and 67.2% have microvascular complications, such as retinopathy, nephropathy, and neuropathy [4]. Among them, diabetic kidney disease (DKD) is a serious diabetes-related complication and is the most important cause of end-stage renal failure [5]. As many deaths occur due to the early onset of cardiovascular disease associated with impaired renal function, the early detection and management of DKD is very important. However, unlike patients with type 1 diabetes, those with type 2 diabetes and impaired renal function may not have albuminuria, and the estimated glomerular filtration rate (eGFR) may be reduced due to various causes, which makes early management of DKD difficult [6]. DKD can be caused by several risk factors, including metabolic syndrome, hypertension, hyperglycemia, insulin resistance, proteinuria, advanced glycation end products (AGEs), and oxidative stress [7]. Therefore, it is necessary to monitor renal function in patients with diabetes using various methods. In particular, patients with metabolic syndrome have an increased risk of type 2 diabetes and a high risk of diabetes-related complications; thus, it is necessary to examine the association between metabolic syndrome and type 2 diabetes [8]. According to the Korean Diabetes Association, approximately 72% of adult patients with diabetes aged 30 years or more had hypercholesterolemia [9], which is closely associated with AGE levels and, compared to those without this disease, is associated with higher blood AGE levels in patients with hyperlipidemia, retinopathy, or peripheral neuropathy [10]. Furthermore, this is closely related to metabolic syndrome. A study that compared skin autofluorescence (SAF) in two patient groups that were grouped according to the presence or absence of metabolic syndrome and measured the AGE level in the skin reported that the SAF value was significantly higher in the metabolic syndrome group (mean SAF: 2.1 AU) compared to the control group (mean SAF: 1.9 AU). Moreover, there was a negative correlation between SAF and high-density lipoprotein cholesterol (HDL-C) [11].

Therefore, to prevent DKD, it is essential to manage metabolic syndrome, which constitutes a cluster of diseases, including hypertension, hyperglycemia, dyslipidemia, and obesity, that occur together, and is one of the major public health problems worldwide [12]. Diabetes was highly correlated with metabolic syndrome even after adjusting for smoking, sex, body mass index, and plasma creatinine [13]. Based on a prospective cohort study, Haffner et al. reported that elevation of blood insulin concentrations preceded metabolic disorders among patients with the insulin resistance syndrome, and this suggested that insulin resistance is the cause of various risk factors for metabolic syndrome [14]; this indicated a close association between metabolic syndrome and diabetes. Therefore, to prevent DKD, it is essential to understand and manage the various causes of metabolic syndrome. In particular, an increased risk is noticeable in middle-aged people (age ≥ 40), which is called the threshold phenomenon, wherein the prevalence of metabolic syndrome, including diabetes, increases rapidly [15]. However, according to data from the Korean Diabetes Association and the Health Insurance Review and Assessment Service, the awareness rate of diabetes among patients with diabetes aged 50 or less is approximately 60%, and that 60.6% of people in their 40s and 60s had no experience with receiving diabetes education. Therefore, it is expected that these individuals may have difficulty controlling diabetes. In addition, poor diabetes management may induce diabetes-related complications, including DKD [15]. According to the Ministry of Health and Welfare in South Korea, the prevalence rates of obesity in individuals in their 40s, 50s, and 60s were 39.0%, 40.2%, and 41.1%, respectively, which indicates that the prevalence of obesity increases steadily with age [16] and that the risk of chronic disease, including metabolic syndrome, is high.

This study aimed to investigate the effects of metabolic syndrome on eGFR in middle-aged participants with diabetes, which is very important for healthy aging, and to provide basic data to enable the development of DKD preventive education programs for middle-aged people.

## 2. Materials and Methods

### 2.1. Study Design and Participants

This cross-sectional descriptive study analyzed data obtained in the 2nd year of the 8th Korea National Health and Nutrition Examination Survey (KNHANES) in 2020 that was conducted by the Korea Disease Control and Prevention Agency. The KNHANES consists of a health survey and health examination at a mobile examination center. In the 2nd year of the 8th KNHANES (2020), data from 3314 households with 7359 participants were recorded, and a health survey and health examination were conducted with 180 primary sample units (PSUs) out of 192 PSUs, and a nutrition survey was conducted with 166 out of 192 PSUs as, compared to the previous year, the number of surveyed households and participants in 2020 decreased by 360 and 750, respectively, due to the coronavirus disease 2019 pandemic. As a heavy metal test was conducted selectively for some of the survey participants, a separate weight was used. The final analysis dataset comprised data from a total of 279 participants aged 40–65 years who were diagnosed with diabetes.

### 2.2. Measures

#### 2.2.1. Demographic and Disease Characteristics and Metabolic Syndrome

The general characteristics of the participants included sex, income level, and education level. The household income level was divided into the lowest, lower middle, upper middle, and highest categories whereas the education level was divided into ≤6, 7–9, 10–12, and ≥13 years categories. Smoking status was assigned according to the current smoking status. Hypertension, dyslipidemia, waist circumference, smoking, blood pressure, fasting blood glucose, lipid analysis, BUN/creatinine, and BMI as risk factors for renal failure were included in the analysis. According to the National Cholesterol Education Program Adult Treatment Panel III (NCEP-ATP III), and the Korean Society for the Study of Obesity, metabolic syndrome is defined as a diagnosis in those meeting three or more of the following five diagnostic criteria: waist circumference ≥90 or ≥85 cm for men and women, respectively; fasting blood sugar ≥100 mg/dL; triglycerides >150 mg, HDL –C level <40 mg/dL for men or <50 mg/dL for women, respectively; and blood pressure ≥130/85 mmHg [17]. Accordingly, data obtained from participants with metabolic syndrome were analyzed.

#### 2.2.2. Biochemical Measurements

Blood samples were collected mainly from the median cubital and cephalic veins after at least 8 h of fasting and, after refrigeration, were sent to a diagnostic medical laboratory on the same day and analyzed within 24 h. Triglycerides, HDL-C, and fasting glucose levels were measured through enzymatic methods on a Hitachi automatic analyzer 7600 (Tokyo, Japan).

#### 2.2.3. Renal Function

The renal function was evaluated based on the eGFR that was calculated using the following estimated glomerular filtration rate: the Modification of Diet in Renal Disease (eGFR–MDRD) formula based on creatinine levels and the individual patient’s age, sex, and ethnicity: eGFR_MDRD_ (mL/min/1.73 m^2^) = 186 × (Scr) − 1.154 × (age) − 0.203 × 0.742 (if female). In accordance with the National Kidney Foundation Kidney Disease Improving Global Outcomes (KDIGO) guideline in the US, renal function was classified as normal (stage 1), mildly decreased (stage 2), mildly to moderately decreased (stage 3a), moderately to severely decreased (stage 3b), and severely decreased (stage 4) (stage 1, eGFR ≥ 90 mL/min/1.73 m^2^; stage 2, 89 to 60 mL/min/1.73 m^2^; stage 3a, 59 to 45 mL/min/1.73 m^2^; stage 3b, 44 to 30 mL/min/1.73 m^2^; and stage 4, <30 mL/min/1.73 m^2^, respectively) [18,19]. To analyze the effects of metabolic syndrome on renal function deterioration in this study, the participants were divided into two groups: a normal renal function group (eGFR ≥ 90 mL/min/1.73 m^2^) and a decreased renal function group (eGFR < 90 mL/min/1.73 m^2^), and factors that potentially affect renal function deterioration were analyzed.

### 2.3. Analytic Strategy

Data were analyzed using SPSS version 22 (IBM Co., Armonk, NY, USA), and the statistical significance level was set at a *p*-value of less than 0.05. With regard to the data from the KNHANES that were used in this study, the KNHANES PSUs were extracted using a multilevel stratified cluster sampling method, a complex sampling design method, to improve the representativeness of the samples and the accuracy of parameter estimation. Data were analyzed using weights in SPSS Complex Samples analysis. The weights used in the KNHANES are multipliers that help represent the entire Korean population, and are calculated by reflecting the extraction rate, response rate, and population distribution. In addition, when a new variable was created by combining several variables, or when a statistical model that simultaneously uses several variables for analysis was fitted, the survey sections, areas, and items of all the variables to be analyzed together were considered to facilitate the selection of appropriate weights. Weights covering a number of survey sections, areas, and items are named as the weights for correlation analysis, and the respective weights by year are provided separately. The general characteristics and differences according to the eGFR stage were analyzed using the Chi-square test, Student’s *t*-test, and ANOVA. The risk factors of metabolic syndrome and the risk of elevated eGFR were analyzed using regression analysis and the correlation between the variables was determined using Pearson’s correlation analysis.

## 3. Results

### 3.1. Demographic and Clinical Characteristics

This study included a total of 279 participants, consisting of 153 men and 126 women. The mean age of the male and female participants was 50.05 and 57.51 years, respectively. With regard to the education level, the highest proportions comprised men who were educated for ≥13 years (35.3%) and women who were educated for 10–12 years (32.5%). Moreover, 131 men (85.6%) and 15 (11.9%) women were smokers, indicating a statistically significant difference between men and women (*p* < 0.01). The prevalence of hypertension and dyslipidemia was 54.2% and 43.7% and 56.2% and 67.5% in men and women, respectively; there was no significant difference between men and women in these parameters (*p* > 0.05). The mean duration of diabetes was 8.77 ± 7.88 and 6.60 ± 5.67 years in men and women, respectively, and was longer in men. The mean waist circumference was 92.71 ± 9.00 and 86.43 ± 8.93 cm in men and women, respectively, and indicated a significant difference between men and women (*p*< 0.01). The systolic blood pressure, a risk factor for metabolic syndrome, was 123.17 ± 14.18 and 121.89 ± 14.87 mmHg in men and women, respectively, without any significant difference between men and women, whereas the diastolic blood pressure was 78.87 ± 9.54 and 74.87 ± 8.19 mmHg in men and women, respectively, which indicated a significant difference between men and women (*p* < 0.01). Moreover, the blood glucose level was 146.98 ± 47.12 and 133.63 ± 37.21 mg/dL and the HDL-C level was 44.15 ± 10.62 and 48.84 ± 11.17 mg/dL in men and women, respectively, whereas the TG level was 180.75 ± 141.39 and 133.7 5 ± 71.84 mg/dL in men and women, respectively, which indicated a significant intergroup difference (*p* < 0.01). The prevalence of metabolic syndrome was 77.8% and 81.0% in men and women, respectively. The mean eGFR was 90.29 ± 15.33 mL/min/1.73 m^2^ in men and 95.16 ± 11.07 mL/min/1.73 m^2^ in women and was significantly higher in women (*p* < 0.01) (Table 1).

### 3.2. eGFR According to the General Characteristics

Based on the eGFR, the cohort was divided into stage 1: normal group (eGFR ≥ 90 mL/min/1.73 m^2^) and stage 2: decreased renal function group (eGFR < 90 mL/min/1.73 m^2^), and the differences were analyzed according to the general characteristics. The proportions of men and women in the decreased renal function group were 68.9% and 31.1%, respectively. The mean age in stage 1 and stage 2 groups was 56.14 and 57.83 years, respectively, indicating that the stage increased with age (*p* < 0.01). The mean duration of diabetes mellitus was 7.13 ± 6.41 and 9.12 ± 8.36 years in the stage 1 and stage 2 groups, respectively, indicating a significant intergroup difference. The mean total cholesterol level was 156.33 ± 34.47 in the stage 1 group, and 169.61 ± 41.73 in the stage 2 group, which indicated a significant intergroup difference (*p* < 0.01). The prevalence of metabolic syndrome was 77.8% in the stage 1 group and 82.2% in the stage 2 group, and there was no significant intergroup difference (*p* > 0.05) (Table 2).

### 3.3. Association for Metabolic Syndrome among the Subitems and the eGFR

The risk factors for metabolic syndromes, such as blood pressure, waist circumference, and levels of fasting blood glucose, triglyceride, HDL-C levels, and eGFR, were analyzed using binary logistic analysis. The results revealed that the OR for metabolic syndrome in men was 1.07 for systolic blood pressure (OR: 1.07, 95% CI: 1.02–1.13) and 1.11 for waist circumference (OR: 1.11, 95% CI: 1.03–1.19) (*p* < 0.05), which indicated that systolic blood pressure and waist circumference were risk factors for metabolic syndrome in men. The odds ratio for metabolic syndrome in women was 1.14 for waist circumference (OR: 1.14, 95% CI: 1.04–1.24) and 0.89 for HDL-C (OR: 0.89, 95% CI: 0.83–0.95), which indicated that waist circumference and HDL-C were risk factors for metabolic syndrome in women (*p* < 0.05) (Table 3).

### 3.4. Correlation of Metabolic Syndrome among the Sub-Items with the eGFR

The analysis of the correlation between eGFR and risk factors for metabolic syndrome showed that systolic blood pressure had a significant positive correlation with the diastolic blood pressure levels, waist circumference, blood glucose levels, and triglycerides, and a significant negative correlation with the eGFR (*p* < 0.05). Diastolic blood pressure had a positive correlation with systolic blood pressure, waist circumference, blood glucose, and triglycerides. Waist circumference had a significant positive correlation with systolic blood pressure, diastolic blood pressure, and triglycerides and a significant negative correlation with HDL-C levels. The blood glucose levels had a significant positive correlation with systolic blood pressure, diastolic blood pressure, and triglycerides, whereas HDL-C levels had a significant negative correlation with waist circumference, blood glucose, and triglyceride levels. In addition, triglycerides showed a significant correlation with systolic blood pressure, diastolic blood pressure, waist circumference, blood glucose, and HDL-C levels, which are factors for metabolic syndrome. The eGFR showed a significant negative correlation with systolic blood pressure (*p* < 0.05) (Table 4).

## 4. Discussion

To identify the factors that affect renal function in middle-aged patients with diabetes, this study analyzed the correlation between metabolic syndrome, its risk factors, and renal function. We found that diastolic blood pressure, fasting blood glucose, triglyceride, and HDL-C levels were higher in men than in women, whereas the eGFR was lower in men than in women. This result is similar to the results of previous studies and indicates that sex differences in renal function can be influenced by sex hormones and sex-related differences in the anatomy of the kidney, stress response, lipid metabolism, and blood pressure [20,21]. In addition, the reasons for the more rapid decrease in renal function in men aged 40–60 years than in women of the same age could be that men have more unhealthy lifestyle habits, such as smoking and alcohol intake, and that testosterone has harmful effects on kidney damage [22,23]. However, in the case of women, estrogen levels have a protective effect on the kidney [24], although women with hyperglycemia due to rapid hormonal changes after middle age showed decreased renal blood flow and increased renal vascular resistance and filtration fraction [25,26,27]. It is thus highly necessary to monitor renal function in middle-aged people with diabetes, regardless of sex.

DKD occurs in approximately 40% of patients with diabetes worldwide, and its early detection and appropriate treatment can slow the progression of chronic kidney disease [28]. In 2017, approximately 34% and 36% of middle-aged men and women with diabetes, respectively, died of DKD worldwide, and this proportion has been increasing since 1990 [29]. Cosmo et al. reported that age, dyslipidemia, etc. served as independent risk factors for metabolic syndrome in patients with diabetes and had a major influence on the early onset of low e-GFR [30]. The results of this study showed that the mean age was higher, the duration of diabetes was longer, and blood cholesterol levels were higher in those with eGFR < 90 mL/min/1.73 m^2^ compared to those with eGFR ≥ 90 mL/min/1.73 m^2^, which concurs with the results of previous studies.

Metabolic syndrome comprises a cluster of cardiovascular risk factors wherein several types of metabolic abnormalities, including central obesity, impaired fasting blood glucose, dyslipidemia, and hypertension, occur together, and is known to increase insulin resistance [31]. In this study, 77.8% and 82.2% of male and female patients with diabetes, respectively, had metabolic syndrome, and this prevalence is very high compared to the 30.4% prevalence of metabolic syndrome in the general population of middle-aged individuals [32]. This is attributed to the fact that both type 2 diabetes and metabolic syndrome have the same pathophysiology of insulin resistance, and more severe insulin resistance is associated with a higher prevalence of metabolic syndrome [33]. However, metabolic syndrome in patients with type 2 diabetes is significantly associated with macro-vascular and micro-vascular complications, including decreased renal function [34,35,36], and, in particular, is highly correlated with decreased renal function in middle-aged people (adults in their 40s and 50s [37]. Therefore, healthcare professionals need to pay attention to the high prevalence of metabolic syndrome in this population.

In the diagnostic criteria for metabolic syndrome, all five diagnostic factors are considered to have the same weight. However, this study found that systolic blood pressure and waist circumference in men, and waist circumference and HDL-C level in women were factors that were associated with a higher risk for metabolic syndrome. According to a study by Boo et al., there was no significant age-related change in the prevalence of dyslipidemia among men, whereas there was a significant age-related change in the prevalence of dyslipidemia among women, which was explained by menopausal hormonal changes in women [38].

A decrease in the estrogen level affects the distribution of body fat, thereby increasing abdominal fat, insulin resistance, and dyslipidemia, and estrogen plays an important role in maintaining adequate HDL-C levels [39]. Lym et al. analyzed metabolic-syndrome-related risk factors in Korean adults in accordance with the ATP III diagnostic criteria, and found that the prevalence of metabolic syndrome was high in men with high blood pressure and high waist circumference, whereas the prevalence of metabolic syndrome was high in women with high blood pressure, high waist circumference, and low HDL-C levels [40]. In particular, the abovementioned study highlighted that waist circumference, that is, abdominal obesity, was a very important risk factor for metabolic syndrome [41], and a similar tendency was identified in this study.

Lee et al. reported that DKD occurred most frequently in the 6 to 8 years after the diagnosis of diabetes [42]. The participants in this study had a high risk of DKD, with a mean disease duration of 7.13 years in men and 9.12 years in women. In particular, an increased duration of diabetes tended to be associated with an increased risk of hyperglycemia and complications of diabetes [43]. Moehlecke et al. reported that renal function and hypertension were the most strongly correlated individual risk components [44]. These results are similar to the results of our study, which showed that the duration of diabetes was relatively longer in those with eGFR < 90 mL/min/1.73 m^2^ compared to the normal group, and that the eGFR had a significant negative correlation with systolic blood pressure. Hypertension is the most important factor for exacerbation of kidney disease in patients with diabetes [31]. In particular, this study found that systolic blood pressure had a significant correlation with diastolic blood pressure, triglyceride levels, waist circumference, and blood glucose levels, which are risk factors for metabolic syndrome. Appropriate management of these factors is thus essential, and it is important for healthcare professionals to recognize the high risk for metabolic syndrome in patients with type 2 diabetes, and to minimize the potential risk of diabetes-induced vascular complications, including renal failure [34].

The limitations of this study include the following drawbacks. A time-series analysis of DKD that requires long-term follow-up data was not performed as this study used public data, and follow-up was not possible. In addition, middle-aged participants with diabetes did not have the same underlying diseases, and the details of the medications they were taking were not recorded in the database. This confers a limitation in that the risk of comorbidities might be underestimated or the degree of control of comorbidities may be unclear. However, this study is significant in that it defined variables based on laboratory test results and estimated the degree of association between eGFR and metabolic syndrome. Therefore, based on the results of this study, customized cohort studies are required for the prevention and follow-up of metabolic syndrome.

## 5. Conclusions

Middle-aged participants with diabetes whose eGFR was <90 mL/min/1.73 m^2^ showed a significant difference in their risk for metabolic syndrome based on sex, age, disease duration, and total cholesterol concentrations. Systolic blood pressure and waist circumference in men, and waist circumference and HDL-C level in women were identified as risk factors that contribute to the increasing prevalence of metabolic syndrome. Diabetes shows an increasing trend worldwide, and it is thus necessary to control and prevent a number of diabetes-related complications, including impaired renal function. In particular, middle age is an important period when the prevalence of chronic diseases increases, and their management is more urgently required. To this end, it is necessary to develop systematic and comprehensive prevention and management measures that are customized to a middle-aged individual’s physical and mental health status to ensure healthy aging.

## Figures and Tables

**Table 1 ijerph-19-11832-t001:** Baseline characteristics of participants (*n* = 279).

Category	Male (*n* = 153)	Female (*n* = 126)	*p*
	N (%), M ± SD	N (%), M ± SD	
Age, years	56.05 ± 6.31	57.51 ± 5.65	0.051
House Income			
Lowest	24 (15.7)	23 (18.3)	0.682
Lower middle	35 (22.9)	33 (26.2)	
Upper middle	43 (28.1)	36 (28.6)	
Highest	51 (33.3)	34 (27.0)	
Education, years			
0–6	14 (9.2)	30 (23.8)	0.001
7–9	20 (13.1)	21 (16.7)	
10–12	48 (31.4)	41 (32.5)	
≥13	54 (35.3)	25 (19.8)	
Smoking			
No	22 (14.4)	111 (88.1)	0.000
Yes	131 (85.6)	15 (11.9)	
Hypertension			
No	70 (45.8)	71 (56.3)	0.078
Yes	83 (54.2)	55 (43.7)	
Dyslipidemia			
No	67 (43.8)	41 (32.5)	0.055
yes	86 (56.2)	85 (67.5)	
Duration of diabetes, years	8.77 ± 7.88	6.60 ± 5.67	0.011
Waist circumference, cm	92.71 ± 9.00	86.43 ± 8.93	0.000
Body mass index, kg/m^2^	25.86 ± 3.63	25.33 ± 3.41	0.221
Systolic blood pressure, mmHg	123.17 ± 14.18	121.89 ± 14.87	0.467
Diastolic blood pressure, mmHg	78.87 ± 9.54	74.87 ± 8.19	0.000
Fasting blood glucose, mg/dL	146.98 ± 47.12	133.63 ± 37.21	0.012
HbA1C, %	7.46 ± 1.30	7.33 ± 1.52	0.465
Triglyceride, mg/d	180.75 ± 141.39	133.75 ± 71.84	0.001
HDL-cholesterol, mg/dL	44.15 ± 10.62	48.84 ± 11.17	0.001
LDL-cholesterol, mg/dL	96.35 ± 32.95	98.15 ± 31.30	0.861
Total cholesterol, mg/dL	146.98 ± 39.49	168.61 ± 40.20	0.191
Blood urea nitrogen, mg/dL	16.38 ± 5.69	15.09 ± 4.61	0.045
Creatinine, mg/dL	0.97 ± 0.60	0.67 ± 0.13	0.000
Albuminuria, mg/dL	70.60 ± 243.85	31.61 ± 118.93	0.114
Metabolic syndrome			
No	30 (19.6)	23 (18.2)	0.717
Yes	119 (77.8)	102 (81.0)	
Unmeasurable	4 (2.6)	1 (0.8)	
eGFR, mL/min/1.73 m^2^	90.29 ± 15.33	95.16 ± 11.07	0.003

Abbreviations: HDL, high-density lipoprotein; LDL, low-density lipoprotein; eGFR, estimated glomerular filtration rate; HbA_1_C, glycosylated hemoglobin A1c.

**Table 2 ijerph-19-11832-t002:** eGFR calculated according to the general characteristics (*n* = 279).

Category	eGFR, mL/min/1.73 m ^2^		*p*
	≥90 (*n* = 189)	<90 (*n* = 90)	
Gender			
Male	91 (48.1)	62 (68.9)	0.001
Female	98 (51.9)	28 (31.1)	
Age, years	56.14 ± 6.29	57.83 ± 5.37	0.029
House Income			
Lowest	30 (15.9)	17 (18.9)	0.411
Lower middle	48 (25.4)	20 (22.2)	
Upper middle	58 (30.7)	21 (23.3)	
Highest	53 (28.0)	32 (35.6)	
Education, years			0.534
0–6	28 (14.8)	16 (17.8)	
7–9	27 (14.3)	14 (15.6)	
10–12	66 (34.9)	23 (25.6)	
≥13	52 (27.5)	27 (30.0)	
Smoking			0.077
No	97 (51.3)	36(40.0)	
Yes	92 (48.7)	54(60.0)	
Hypertension			0.901
No	96 (50.8)	45 (50.0)	
Yes	93 (49.2)	45 (50.0)	
Dyslipidemia			0.125
No	79 (41.8)	29 (32.2)	
Yes	110 (58.2)	61 (67.8)	
Duration of diabetes, years	7.13 ± 6.41	9.12 ± 8.36	0.024
Waist circumference, cm	89.38 ± 8.74	90.93 ± 10.88	0.206
Body mass index, kg/m^2^	25.68 ± 3.41	25.47 ± 3.82	0.649
Systolic blood pressure, mmHg	121.68 ± 13.64	124.52 ± 16.07	0.132
Diastolic blood pressure, mmHg	76.59 ± 9.12	78.10 ± 9.06	0.203
Fasting blood glucose, mg/dL	140.29 ± 41.53	141.97 ± 46.81	0.766
HbA1C, mg/dL	7.39 ± 1.42	7.42 ± 1.38	0.890
Triglyceride, mg/dL	161.48 ± 112.69	154.62 ± 125.64	0.651
HDL-cholesterol, mg/dL	46.81 ± 11.11	45.29 ± 11.08	0.289
LDL-cholesterol, mg/dL	99.63 ± 31.39	90.68 ± 34.30	0.325
Total cholesterol, mg/dL	156.33 ± 34.47	169.61 ± 41.73	0.010
Metabolic syndrome			0.646
No	37 (19.6)	16 (17.8)	
Yes	147 (77.8)	74 (82.2)	
Unmeasurable	2 (2.6)		

Data are presented as mean (±standard deviation) and proportion (percent); Abbreviations: HDL, high-density lipoprotein; LDL, low-density lipoprotein; eGFR, estimated glomerular filtration rate; HbA_1_C, hemoglobin A_1_C.

**Table 3 ijerph-19-11832-t003:** Univariate analysis for metabolic syndrome among the sub-items (BP, WC, FBS, TG, and HDL-C) and eGFR.

Category	Male		Female	
	OR (CI)	*p*	OR (CI)	*p*
eGFR, mL/min/1.73 m^2^	0.99 (0.96–1.02)	0.617	1.03 (0.98–1.09)	0.260
Systolic blood pressure, mmHg	1.07 (1.02–1.13)	0.008	10.36 (0.99–1.09)	0.163
Diastolic blood pressure, mmHg	1.00 (0.93–1.08)	0.998	0.96 (0.88–1.04)	0.314
Waist circumference, cm	1.11 (1.03–1.19)	0.004	1.14 (1.04–1.24)	0.005
Fasting blood glucose, mg/dL	1.00 (0.99–1.01)	0.839	0.99 (0.98–1.01)	0.435
Triglyceride, mg/dL	1.00 (1.00–1.01)	0.383	1.01 (0.99–1.02)	0.390
HDL-cholesterol, mg/dL	0.96 (0.91–1.00)	0.060	0.89 (0.83–0.95)	0.001

Abbreviations: OR, odds ratio; CI, confidence interval; BP, blood pressure; WC, waist circumference; FBS, fasting blood sugar; TG, triglyceride; HDL, high-density lipoprotein; GFR, estimated glomerular filtration rate.

**Table 4 ijerph-19-11832-t004:** Correlation of metabolic syndrome sub-items with eGFR.

	BPsys, mmHg	BPdia, mmHg	WC	FBS	HDL-C	TG
BPsys, mmHg	1					
BPdia, mmHg	0.55 (0.000)	1				
WC, cm	0.26 (0.000)	0.37 (0.000)	1			
FBS, mg/dL	0.21 (0.001)	0.20 (0.001)	0.12 (0.059)	1		
HDL-C, mg/dL	0.03 (0.687)	−0.09 (0.144)	−0.19 (0.002)	−0.04 (0.482)	1	
TG, mg/dL	0.16 (0.008)	0.25 (0.000)	0.12 (0.049)	0.29 (0.000)	−0.30 (0.000)	1
eGFR, mL/min/1.73 m^2^	0.12 (0.049)	0.02 (0.729)	0.02 (0.719)	0.02 (0.809)	0.09 (0.165)	0.07 (0.225)

Abbreviation: eGFR, estimated glomerular filtration rate; BPsys, systolic blood pressure; BPdia; diastolic blood pressure; WC, waist circumference; FBS, fasting blood glucose; HDL-C, high-density lipoprotein cholesterol; TG, Triglyceride; GFR, estimated glomerular filtration rate.

## Data Availability

The data can be found here: http://knhanes.cdc.go.kr (accessed on 4 July 2022).

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
