# Peer review of "Analysis of the Association between Metabolic Syndrome and Renal Function in Middle-Aged Patients with Diabetes"

_ijerph, 2022, doi:10.3390/ijerph191811832_

Round 1
Reviewer 1 Report
1. This manuscript brings up an important topic and provides relevant results. It is scientifically sound, clear, and presented in a well-structured manner. The article is of interest to a broad audience, even though the obtained results are not surprising and were anticipated. The paper is relevant for the field. However, there are still some minor remarks.
2. When writing about the global burden of diabetes, the authors should refer to the current, rather than old sources. On the 1st page, they write “the prevalence of diabetes is gradually increasing and is estimated to reach approximately 590 million by 2035” ([3] Global estimates of diabetes prevalence for 2013 and projections for 2035. Diabetes Res Clin Pract. 2014). Meanwhile, the estimated prevalence is 537 million (10.5%) today with an estimated increase of 643 million people by 2030 (11.3% of the population) with a jump to a staggering 783 million (12.2%) by 2045 (IDF Diabetes Atlas 2021, 10th Ed, www.diabetesatlas.org.).
3. There are dimensionless quantities, such as various ratios, OR, etc. The GFR (throughout the article) and BMI (Table 2) are not applicable to them. They are measured in ml/min/1.73m2 and kg/m2 respectively. The authors omit the units of measurement not only for these values, but also for others in describing the results. That is not entirely accurate. All units of measurement must be clearly indicated. As for the frequently mentioned GFR, it is necessary to state in the text (for example, in methods), that > or < 90 is measured in ml/min/1.73 m2.
4. As a result, in spite of these minor comments, the article may be recommended for publication.
Author Response
Thank you for your sincere review. I replied to your careful comment.
Please check the attached file.

Reviewer 2 Report
I am very impressed with the presented publication, which is extremely interesting and written in a very good style.
The study proposed by the authors presents an interesting area of health and medicine news, which causes interest for both scientists and practitioners.
The authors, no doubt, did a good job, including the application of modern methods and statistic studies in this research. The undoubted advantage of the manuscript is the very specific goal and really interesting and reliable introduction of the presented research. However, while reading the article, only one remark arised, by answering which the authors will improve the presentation of the results.
1. Point 2.2.3: Please mention what the abbreviation MDRD stands for (Modification of Diet in Renal Disease)
Author Response
Thank you very much for your sincere review.
Please see the attached file for the response to your review.